# Kinetics of Torque Teno Virus Viral Load Is Associated with Infection and De Novo Donor Specific Antibodies in the First Year after Kidney Transplantation: A Prospective Cohort Study

**DOI:** 10.3390/v15071464

**Published:** 2023-06-28

**Authors:** Sara Querido, Catarina Martins, Perpétua Gomes, Maria Ana Pessanha, Maria Jorge Arroz, Teresa Adragão, Ana Casqueiro, Regina Oliveira, Inês Costa, Jorge Azinheira, Paulo Paixão, André Weigert

**Affiliations:** 1Renal Transplantation Unit, Nephrology Department, Hospital de Santa Cruz, Centro Hospitalar de Lisboa Ocidental, 2790-134 Carnaxide, Portugal; tadragao@gmail.com (T.A.); ana.salvadinha@gmail.com (A.C.); mroliveira@chlo.min-saude.pt (R.O.); alweigert@chlo.min-saude.pt (A.W.); 2Infection, Sepsis & Antibiotics Resistance Research Group, CHRC—Comprehensive Health Research Center, NOVA Medical School, Faculdade de Ciências Médicas (NMS|FCM), Universidade NOVA de Lisboa, 1150-082 Lisboa, Portugal; paulo.paixao@nms.unl.pt; 3Immune Dysregulation from Pregnancy to Adulthood Research Group, CHRC—Comprehensive Health Research Center, NOVA Medical School, Faculdade de Ciências Médicas (NMS|FCM), Universidade NOVA de Lisboa, 1150-082 Lisboa, Portugal; catarina.martins@nms.unl.pt; 4Laboratory of Clinical Microbiology and Molecular Biology, Department of Clinical Pathology, Centro Hospitalar de Lisboa Ocidental, 1349-019 Lisboa, Portugal; gomes.perpetua@gmail.com (P.G.); mariaana.pessanha@gmail.com (M.A.P.); icosta@chlo.min-saude.pt (I.C.); 5Centro de Investigação Interdisciplinar Egas Moniz (CiiEM), IUEM, 2829-511 Almada, Portugal; 6Flow Cytometry Laboratory, Department of Clinical Pathology, Centro Hospitalar de Lisboa Ocidental, 1349-019 Lisboa, Portugal; marroz@chlo.min-saude.pt; 7Laboratory of Biochemistry, Department of Clinical Pathology, Centro Hospitalar de Lisboa Ocidental, 1349-019 Lisboa, Portugal; jazinheira@chlo.min-saude.pt; 8Pharmacology and Neurosciences Institute, Faculdade de Medicina, Universidade de Lisboa, 1649-004 Lisboa, Portugal

**Keywords:** Torquetenovirus, immunosuppression, kidney transplant, infection, donor-specific antibodies

## Abstract

Torque teno virus (TTV) was recently identified as a potential biomarker for the degree of immunosuppression, and potentially as a predictor of rejection and infection in solid organ transplant patients. We evaluated TTV viral load in kidney transplant (KT) patients during the first year post-transplant to examine overall kinetics and their relationships with deleterious events, including episodes of infection and the formation of de novo donor-specific antibodies (DSAs). In a single-center, prospective observational cohort study, 81 KT patients were monitored at baseline, week 1, and month 1, 3, 6, 9 and 12, post-KT, and whenever required by clinical events. Kidney function, plasma TTV load, immunoglobulins and lymphocyte subpopulations were assessed at each time point. Twenty-six patients (32.1%) presented a total of 38 infection episodes post-KT. Induction immunosuppression with thymoglobulin, compared to basiliximab, was not associated with more infections (*p* = 0.8093). Patients with infectious events had lower T-cells (*p* = 0.0500), CD8^+^ T-cells (*p* = 0.0313) and B-cells (*p* = 0.0009) 1 month post-KT, compared to infection-free patients. Patients with infection also showed higher increases in TTV viral loads between week 1- month 1, post-KT, with TTV viral load variations >2.65 log_10_ cp/mL predicting the development of infectious events during the 12-month study period (*p* < 0.0001; sensitivity 99.73%; specificity 83.67%). Patients who developed de novo DSAs had lower TTV DNA viral loads at month 12 after KT, compared to patients who did not develop DSA (3.7 vs. 5.3 log_10_ cp/mL, *p* = 0.0023). Briefly, evaluating early TTV viremia is a promising strategy for defining infectious risk in the 1st year post-KT. The availability of standardized commercial real-time PCR assays is crucial to further validate this as an effective tool guiding immunosuppression prescription.

## 1. Introduction

Prevention, diagnosis and treatment of infection and rejection are key goals in the care of kidney transplant (KT) patients. Until recently, no reliable biomarker has definitively emerged to define the level of immune function of KT patients. Clinically routine doses of immunosuppressive drugs are mainly guided by the quantification of the calcineurin or mTOR inhibitor trough drug level in peripheral blood, which correlates more closely with the risk of drug-related toxicity than with the effectiveness of immunosuppression [1].

Torque teno virus (TTV) is a small, non-enveloped, circular, single-stranded DNA anellovirus that has recently gained attention as a potential surrogate marker of the net state of immunosuppression [2].

TTV has significant genomic variability due to inter or intragenomic rearrangements; according to the 2018 International Committee on Taxonomy of Viruses (ICTV) classification; in total, 29 species classified in five genogroups are known [2]. Co-infections with several microbial species are extremely frequent (>70%).

TTV can be detected in up to 90% of healthy individuals and it has not been associated with any specific disease, since the adaptive cellular immune responses control TTV infection [3]. However, in KT patients, TTV becomes detectable in up to 100% of patients and is unaffected by conventional antiviral prophylactic drugs [4].

Recent studies evaluated whether peripheral blood levels of TTV might reflect the overall strength of innate and specific immunity [5]. Hence, quantification of TTV viral load and/or TTV kinetics after KT could be a predictive biomarker for the risk of rejection and infection in solid organ transplant patients [2,5,6].

Evidence suggests that high or increasing TTV DNA levels correspond to over-immunosuppression, preceding the occurrence of infectious complications after KT, whereas low or decreasing viral loads correspond to under-immunosuppression, signaling a high risk for the development of acute rejection [7,8].

In clinical practice, TTV viral load might potentially be used to predict increased risk of both rejection and infection, and, thus, it could be a tool to use in the planning of customized immunosuppression strategies. Nevertheless, the ideal threshold for reduction of immunosuppression and the best time points to measure TTV viremia in order to modulate immunosuppression are yet to be determined.

Monitoring cell-mediated immunity (CMI) has been proposed as a promising strategy to reduce the incidence of post-transplant infection by individualizing immunosuppressive therapy. Analysis of CMI status is expensive and cumbersome, and surrogate parameters, such as peripheral blood lymphocyte subpopulations (PBLSs), might be an efficient alternative method of evaluation. Thus, the kinetics of PBLSs could be a helpful tool in the identification of recipients at risk of post-transplant infection [9,10].

In addition, measurement of serum immunoglobulins levels is a widely available and affordable surrogate marker for the functional status of humoral immunity. Indeed, the presence of post-transplant hypogammaglobinemia (HGG) has been shown to be associated with an increased risk of infection after KT [11].

In this study, we determined plasma TTV load kinetics, serum immunoglobulins and PBLSs from 81 de novo KT patients, drawn before and at different time points after KT, and explored their association with the development of infection and the formation of de novo donor-specific antibodies (DSAs) after KT. A joint model was built to analyze these longitudinal endpoints with the repeated TTV load measurements. Correlation of TTV loads against clinical endpoints corroborates TTV as a potential biomarker of functional immunity.

## 2. Material and Methods

### 2.1. Study Design and Population

This single-center prospective observational cohort study, included all 92 consecutive adult (≥18 years of age) recipients of a kidney allograft at Hospital de Santa Cruz, Portugal, between the 1st of February 2019 and the 28th of February 2021. Patients were followed at the outpatient clinic of Hospital de Santa Cruz for 12 months after KT. Patients who ended the follow-up before the first year after KT due to change of the outpatient care center (*n* = 6), graft-loss (*n* = 4) or death (*n* = 1) were excluded from the analysis. Finally, 81 patients were included in the cohort analysis. All participants provided informed consent.

Scheduled follow-up clinical and laboratory evaluations were carried out at baseline, in the first week and in the 1st, 3rd, 6th, 9th and 12th months after KT, and whenever required by clinical events. Clinical indications for additional evaluations included infectious events or biopsy-proven graft rejection.

Infectious events were defined as any bacterial, fungal or viral infection requiring antimicrobial or antiviral therapy and in need of hospitalization or prolongation of a hospital stay.

Polyomavirus infections were defined according to recommendations of the Banff Working Group and the American Society of Transplantation Infectious Diseases Community of Practice guidelines [12,13]. Briefly, plasma BKPyV viral load ≥ 1 × 10^4^ copies/milliliter (cp/mL) was defined as presumptive polyomavirus nephropathy (pPVAN) and polyomavirus nephropathy (PVAN) was defined by biopsy.

Cytomegalovirus (CMV) viremia was defined by CMV replication in plasma [14].

Biopsy-proven acute rejection episodes were classified according to the 2019 update of the Banff classification [15].

The study is in compliance with the Declaration of Helsinki, follows national and international guidelines for health data protection and was approved by the Ethics Committee of the “Centro Hospitalar de Lisboa Ocidental” (approval number 20170700050).

### 2.2. Data Collection

Demographic characteristics (age, gender), type of donation (living/deceased donor), induction and maintenance immunosuppression and immunologic risk profile (number of mismatches between donor and recipient and presence of class I and class II anti-HLA antibodies) were collected at baseline.

Scheduled laboratory data collection included determination of complete blood count, C-reactive protein, immunoglobulins (IgG, IgA, IgM), complement (C3, C4), lymphocyte absolute counts and subsets (including total T cells, CD4+ T cells, CD8+ T cells, B cells and NK cells), BK polyomavirus (BKPyV) viral load in plasma and urine, JC polyomavirus (JCPyV) viral load in plasma and urine, TTV and CMV viral load in plasma, estimated glomerular filtration rate (eGFR) and HLA class I and II antibodies (including median fluorescence intensity (MFI) levels of donor-specific antibodies; a threshold of 1.000 MFI was considered positive).

Estimated glomerular filtration rate (eGFR) was calculated by the Chronic Kidney Disease Epidemiology Collaboration (CKD-EPI) equation [16].

### 2.3. TTV Analysis

Quantitative TTV DNA viral load extraction from plasma samples and amplification of DNA was performed as previously described [17,18]. In brief, DNA extraction was carried out using the eMAG System (BioMerieux, Marcy, France). For DNA amplification and quantification, the Argene R-Gene TTV quantification kit (BioMerieux) was used on an Applied Biosystems 7500 (Thermofisher, Waltham, MA, USA) according to the manufacturer’s instructions. The R-Gene assay is designed to detect the majority of TTV genotypes (1, 6, 8, 10, 12, 15, 16, 19, 27, 28). The threshold defining positivity was 100 copies/mL, as defined by manufacturer. Results are expressed in log_10_ copies/mL.

### 2.4. BKPyV and JCPyV Analysis

The presence of BKPyV and JCPyV were assessed in urine (viruria) and in plasma samples (viremia). For the detection of JCPyV and BKPyV, we used the commercial assays JCPyV ELITe MGB^®^ Kit and BKPyV ELITe MGB^®^ Kit. These assays are CE-IVD validated on diverse range of sample types, in combination with ELITe InGenius^®^, a fully automated sample-to-result solution. The primers and the JCPyV and BKPyV-specific probe (stabilized by MGB^®^ group, labelled by FAM fluorophore and quenched by a non-fluorescent molecule) are specific for the Large T antigen region of the JCPyV gene and the Large T antigen gene of BKPyV. The primers and the probe for the internal control (stabilized with MGB^®^ group, labelled by AP525 fluorophore, analogous to VIC, and quenched by a non-fluorescent molecule) are specific for the artificial DNA sequence.

The sample volume used to extract DNA was 200 µL. In both assays, two amplification reactions were performed, starting from extracted DNA. For BKPyV, a specific primer for the region of the Large T antigen gene of BKPyV and a specific primer for the region of the human beta-globin gene (internal control) were used. For JCPyV, a specific primer for the Large T antigen region of the JCPyV gene and a specific primer for an artificial sequence of DNA (internal control) were used. BKV and JCPyV-specific probes with ELITE MGB^®^ technology, labelled with FAM fluorophore, is activated when it hybridizes with the specific product of the BKPyV and JCPyV amplification reaction. Viral load is obtained, in each case, through a calibration curve. The threshold defining positivity for BKPyV was 165 copies/mL in plasma and 89 copies/mL in urine. For JCPyV, both plasma and urine threshold were 500 copies/mL. Results are expressed in log_10_ copies/mL.

Detection of BKPyV viremia in two consecutive analyses, triggered discontinuation of the antimetabolite and initiation of mTOR inhibitor (everolimus; target 12-h trough levels of 3–7 ng/mL), as previously published by our group [19]. Calcineurin inhibitor target trough levels were also reduced (tacrolimus target 12-h trough levels of 3–5 ng/mL), in accordance to clinical practice in our center. Prednisolone was kept at 2.5 to 5 mg qday.

### 2.5. CMV Analysis

CMV viremia was assessed in plasma samples. Briefly, after DNA extraction two amplification reactions are performed: a specific reaction for the exon 4 region of the CMV MIEA gene (major immediate early antigen, HCMVUL123) and a specific reaction for a region of the human beta-globin gene (internal control of inhibition). The CMV-specific probe with ELITe MGB^®^ technology, labelled with FAM fluorophore, is activated when hybridizes with the specific product of the CMV amplification reaction. The internal control-specific probe with ELITe MGB^®^ technology, labelled with AP525 fluorophore (analogous to VIC), is activated when it hybridizes with the specific product of the internal control amplification reaction. Viral load is obtained through a calibration curve. The threshold defining positivity was 178 copies/mL. Results are expressed in log_10_ copies/mL.

### 2.6. Lymphocyte Subsets

Mature human lymphocyte subsets in peripheral whole blood were evaluated by multiparametric flow cytometry: T lymphocytes (CD3+), B lymphocytes (CD19+), Natural killer (NK) lymphocytes (CD3–CD16+ and/or CD56+), Helper/inducer T lymphocytes (CD3+CD4+) and Suppressor/cytotoxic T lymphocytes (CD3+CD8+).

The BD Multitest™ 6-Color TBNK kit which contains FITC-labeled CD3, clone SK7; PE-labeled CD16, clone B73.1 and PE-labeled CD56, clone NCAM16.2; PerCP-Cy 5.5–labeled CD45, clone 2D1 (HLe-1); PE-Cy7–labeled CD4, clone SK3; APC-labeled CD19, clone SJ25C1 and APC-Cy7–labeled CD8, clone SK1 was used with BD Trucount™ Tubes.

After sample incubation, a specific lyse/no wash procedure followed, and the cells were acquired using a BD FACSCanto™ II flow cytometer. Using cytometer-specific BD FACSCanto^TM^ software (version 3.1), the results of the different cell subsets were obtained as a percentage of lymphocytes and the number of positive cells per microliter of blood (absolute count).

### 2.7. Immunosuppressive Protocols

KT recipients received basiliximab or antithymocyte globulin as induction therapy, except if HLAs were identical (*n* = 1), in which case, no induction therapy was used. Basiliximab (20 mg IV) was administered in the first and fourth day after KT. Antithymocyte globulin (1.25 mg/kg/day IV) was administered since the first day and optimally until the seventh day after KT. Methylprednisolone (500 mg on 1st day, 250 mg on 2nd day, 125 mg on 3rd, and 80 mg on 4th day IV after KT) was included in all immunosuppressive induction protocols. The choice of the immunosuppressive regimen depended mainly on patient’s immunologic profile (% of panel reactive antibodies, number of HLA mismatches with the donor, preformed donor-specific antibodies), although delayed graft function led to the use of antithymocyte globulin and delayed introduction of calcineurin inhibitor. Rituximab (375 mg/m^2^ for 2 doses 2 weeks apart) were used as induction therapy in addition to thymoglobulin in two highly sensitized patients. Initial maintenance immunosuppressive therapy included tacrolimus, mycophenolate mofetil, and prednisone. Tacrolimus was administered orally at 0.15 mg/kg/day divided in two doses and adjusted to maintain a target trough concentration between 4 and 10 ng/mL, depending on the time elapsed after KT. Prednisolone was prescribed since the fifth day after KT (0.6 mg/kg) and was tapered to 5 mg/day during the first 3 months after KT. Mycophenolate mofetil (1000 mg orally twice daily) was started after KT and was reduced if adverse events appeared, otherwise it was reduced to 1000 to 1500 mg daily dose after the first 3–6 months.

### 2.8. Kidney Transplant Biopsies

No surveillance or protocol biopsies were performed.

Acute cellular rejection (ACR) was treated with methylprednisolone (500 mg/d IV) for 3 days. Antibody-mediated rejection (AMR) was treated with a variable combination of plasmapheresis, intravenous immunoglobulin and/or rituximab.

All subjects with a rise in creatinine who underwent indication biopsies were simultaneously assessed for BKPyV and JCPyV viremia at the time of biopsy. The diagnosis of proven PVAN was sought by demonstrating cytopathic changes of tubular epithelial cells in the allograft tissue and confirmed by immunohistochemistry or in situ hybridization.

### 2.9. Prophylaxis Protocols

All KT recipients received trimethoprim/sulfamethoxazole (480 mg qd) or atovaquone 750 mg bid as *Pneumocystis jirovecii* pneumonia prophylaxis for 1 year. Valganciclovir (900 mg qd, adjusted to kidney function) was given to patients which induction therapy included antithymocyte globulin and/or rituximab or in receptor CMV IgG-negative/donor CMV IgG-positive pairs for 6 months.

### 2.10. Statistical Analysis

GraphPad Prism software, version 9.5.1, was used for statistical analysis (Graph Pad, San Diego, CA, USA). For categorical variables, group comparisons were assessed using the Fisher’s exact test or the Chi Square test, as applicable. For the paired group analysis through time, the time point values for continuous variables were assessed by ANOVA mixed-effects analysis, with the Geisser–Greenhouse correction, followed by the Tukey’s multiple comparisons test, with individual variances computed for each comparison.

Furthermore, when patients were divided according to their distinct characteristics, continuous variables of different subgroups were compared using the nonparametric Mann–Whitney U-test. For the parameters with statistically significant differences between subgroups, receiver operating characteristics (ROC) curve and area under the curve (AUC), as well as sensitivity and specificity were also calculated to assess their statistical accuracy for distinguishing between patients with and without risk to develop infections complications or de novo specific anti-donor antibodies.

Finally, correlation studies were assessed by the nonparametric Spearman correlation coefficient. Significance was considered for *p* < 0.05.

## 3. Results

### 3.1. Patients’ Characteristics

A total of 81 patients were included in the cohort analysis. Median age at transplantation was 52.0 [42.5; 61.5] years; 65.4% were male; 12.4% had a previous KT; 13.6% received a kidney from a living donor; 53.1% of patients received T-cell depleting as induction therapy, 24.7% of patients had pre-formed DSAs and 13.6% developed de novo DSAs in the first year after KT. Clinical and demographical data of the study population are detailed in Table 1.

Regarding the occurrence of study outcomes, 26 patients (32.1%) presented a total of 38 episodes of infection after KT (incidence rate: 1.32 episodes per 1000 transplant-days). Details on the affected organ systems and causative pathogens are described in Table 2.

Among the 81 patients, four (4.9%) had a biopsy-proven cellular (*n* = 2) or antibody-mediated acute rejection (*n* = 2), 17.2% (*n* = 14) had a presumptive PVAN and 1 patient (1.2%) had a biopsy-proven PVAN.

### 3.2. Dynamics of Immune and Microbiologic Parameters within the 1st Year after KT

Kinetics of immune and microbiologic parameters are detailed in Appendix A.

TTV DNA viral load was available before KT for 80 patients. Twenty-two recipients (27.5%) had undetectable TTV viral load at baseline; however, 79 became TTV-positive after KT. A progressive increase in TTV viral load was observed from baseline (3.10 log_10_ [0; 4.28] cp/mL) to peak at month 3 (7.2 log_10_ [5.8; 8.2] cp/mL) and a slight decrease was seen after month 6 (6.1 log_10_ [4.5; 7.8] cp/mL) (*p* < 0.0001 for the paired comparison of all time points, ANOVA). No patient had detectable BKPyV and JCPyV viremia at baseline.

Kinetics of log TTV DNA during the first year after KT is detailed in Figure 1.

Regarding immunological parameters, significant changes in the kinetics of complement C3, complement C4, immunoglobulin G (IgG), immunoglobulin A (IgA), immunoglobulin M (IgM), total white blood count (*p* < 0.0001), total lymphocytes count (*p* = 0.0039), CD3+ T cells (*p* = 0.0004), CD4+ T cells (*p* = 0.0128), CD8+ T cells (*p* < 0.0001), CD 19+ B cells (*p* < 0.0001) and natural killer cells (*p* < 0.0001) were detected during the first year after KT. (Figure 1/Appendix A).

More pronounced changes were found for IgG, IgA and IgM, since all three immunoglobulins abruptly decreased in the first week after KT. Levels recovered after the first month, but never reached baseline values during the study.

Kinetics of tacrolimus levels (*p* < 0.0001) and eGFR (*p* < 0.0001) varied throughout the year after KT. For JCPyV and CMV, the number of patients with detectable viremia did not change significantly along the study (*p* = NS). However, the number of BKPyV-positive patients increased significantly in urine after the 1st month (*p* = 0.0001) and in plasma after the 3rd month. (*p* = 0.0016).

Regarding polyomavirus viruria, JCPyV viruria appeared earlier than BKPyV viruria, as almost 19% of patients had JCPyV viruria within the first week after KT. Furthermore, at 6 months after KT, prevalence of viruria for both viruses was similar (BKPyV viruria: 30.00%, JCPyV viruria: 31.25%). Polyomavirus viremia was more prevalent for BKPyV than for JCPyV, and reached a peak in the 3rd month after KT, when 14.81% of patients had BKPyV detected in urine. No differences were found in JCV and BKPyV prevalence between patients with and without infectious events.

### 3.3. Characteristics of Patients Admitted Due to Infectious Events after KT

Within the first year after KT, almost one-third of patients were admitted due to an infectious event (*n* = 26; 32%). Thus, we further characterized their immune and microbiological background and evolution to determine the potential utility of these markers in stratification of patients based on their risk of infection. For this purpose, we compared patients who were admitted in the first year after KT due to an infectious event to patients without any clinically relevant infection in the same time period (*n* = 49). Six patients were also admitted along the study follow-up period for non-infectious-related conditions; thus, they were not included in this comparison.

Overall, no differences between patients with and without infectious events were found in induction immunosuppression (thymoglobulin: 53.8% vs. 49.0%, respectively, *p* = 0.8093), age (54.0 [45.0; 63.5] vs. 49.0 [43.0; 62.0] years, respectively, *p* = 0.4234), gender (female: 38.5% vs. 36.8%, respectively, *p* > 0.9999) or presence of ureteral stent (23.1% vs. 18.4%, respectively, *p* = 0.7626). The infectious event leading to admission occurred about 2.9 ± 2.2 months after transplant (seven patients < 1 month after KT, eight patients between months 1 and 3 after KT, nine patients between months 3 and 6 after KT and two patients >6 months after KT).

Clinical and laboratory parameters of both groups are detailed in Table 3.

Considering the important differences observed throughout the study follow-up, we further looked at the preceding individual parameters in each subgroup of patients with subsequent infectious-related admissions (i.e., dynamics from baseline to the end of the 1st week in patients suffering infectious events up to the end of the 1st month post KT) (Table 4).

### 3.4. Infectious Events within the 1st Month after KT

During the first month after KT, seven patients were admitted due to an infectious event (four with a viral and three with a bacterial infection). The proportion of patients with admission due to infection who had received intravenous human immunoglobulin (IVIg) as part of induction immunosuppression was higher than the proportion among patients free of infectious events in the first month (43.9% vs. 10.2%, respectively, *p* = 0.05). No differences were observed for the baseline characteristics, although there were higher levels of IgM in the infection group at the 1st week follow-up (*p* = 0.0576). At the month 1 assessment, compared to infection-free patients, patients with infectious events had lower CD3+ T cells (1093 vs. 498 cells/µL, respectively, *p* = 0.05), lower CD8+ T cells (363 vs. 128 cells/µL, respectively, *p* = 0.0313) and lower CD19+ B cells (234 vs. 33 cells/µL, respectively, *p* = 0.0009).

All 26 patients who needed hospitalization due to infection at any time during the study follow-up after KT had higher increases in TTV viral loads between the 1st week and the 1st month assessment than did those patients without infectious events (4.65 [3.6–5.7] vs. 1.3 [0.5–2.4] log_10_ cp/mL, respectively, *p* < 0.0001).

The evaluation of this variation as a biomarker included using an ROC curve analysis and a simple logistic regression model. We performed a sensitivity/specificity analysis to identify a variation cut-off, measured as an increase in log TTV viral load between week 1 and month 1 that is associated with an with a greater probability of developing an infection episode. We found that for each 1 log TTV viral load increase, the likelihood of acquiring an infection increases 4.182-fold (OR: 4.182; 95% CI: 2.353–9.814; *p* ≤ 0.0001). A variation of TTV viral load between week 1 and month 1 > 2.65 log_10_ cp/mL was determined as a cut-off for considering the development of an infectious event within 12 months (*p* < 0.0001), with a sensitivity of 99.73% and a specificity of 83.67% for discriminating an infectious event.

### 3.5. Infectious Events between the 1st and 3rd Months after KT

Eight patients developed an infectious event between the 1st and 3rd months after KT (six bacterial and two viral infections).

Of note, patients with episodes of infection in this time point had higher variations in TTV viral loads between the 1st week and the 1st month after KT, compared to those remaining free from this complication (4.65 [3.6–5.7] vs. 1.3 [0.5–2.4] log_10_ cp/mL, respectively, *p* < 0.0001). No differences were found either in TTV viral load or in TTV kinetics between patients with and without infectious events during this specific time interval.

### 3.6. Infectious Events between the 3rd and 6th Months after KT

Nine patients were diagnosed with an infection episode between 3rd and 6th months after KT. Patients with infectious events had lower CD3+ T cells (540 vs. 994 cells/µL, *p* = 0.05) and lower CD4+ T cells (161 vs. 605 cells/µL, *p* = 0.0029), when compared to infection-free patients in the same time interval. No changes were found either in TTV viral load or in TTV kinetics between patients with and without infectious events during this specific time point.

### 3.7. Characteristics of Patients with Preformed Donor Specific Antibodies

Twenty patients (24.7%) had formed donor-specific antibodies (DSAs) before KT. As expected, patients with preformed DSAs were more likely to receive thymoglobulin for induction immunosuppression than were patients without DSAs before KT (75.0% vs. 45.9%, respectively, *p* = 0.0236). One of these patients also received rituximab.

Moreover, patients with preformed DSAs had higher TTV DNA viral loads at the 6th (7.45 vs. 5.55 log_10_ cp/mL, *p* = 0.0133) and 9th months (6.70 vs. 5.20 log_10_ cp/mL, *p* = 0.0107), when compared to patients without preformed antibodies.

Regarding immunoglobulin data, those patients with preformed DSAs were more likely to receive IVIg as part of induction immunosuppression (35.0% vs. 1.6% for patients without pre-formed DSAs, *p* < 0.0001). As expected, IgG levels were higher in previously sensitized patients in the 1st week (1051.0 vs. 818.0 mg/dL, *p* = 0.0263) and 1st month (1005.0 vs. 784.5 mg/dL, *p* = 0.0552) after KT than in patients without pre formed DSAs, since more of the first group of patients received IVIg as part of induction therapy.

Considering IgM, this immunoglobulin was lower at all time points after KT for patients with preformed DSAs compared to patients without those antibodies. This difference reached statistical significance at the 1st week (43.5 vs. 68.0 mg/dL, respectively, *p* = 0.049) and 3rd, (42.0 vs. 72.0 mg/dL, respectively, *p* = 0.0245) 6th (50.5 vs. 71.5 mg/dL, respectively, *p* = 0.0566) and 9th month (49.0 vs. 82.0 cp/mL, respectively, *p* = 0.0145) follow-up assessment after KT.

### 3.8. Characteristics of Patients with De Novo Donor Specific Antibodies after KT

Sixty-one patients were DSA-naïve before KT; however, 11 of these patients (18.03%) developed de novo DSAs after KT. Patients with de novo DSAs were younger than patients without DSAs within the study period (46.55 vs. 55.33 years, respectively, *p* = 0.0485) (Table 5).

Between patients with and without de novo DSAs detected along follow up, no differences were found in immunosuppressive induction protocols (thymoglobulin: 63.6% vs. 52%, respectively, *p* = 0.5258) or in the rate of acute rejection (9.09% vs. 2.00%, respectively, *p* = 0.3306).

Regarding TTV viral load, patients who developed de novo DSAs only showed lower TTV DNA viral loads in the 12th month after KT compared to patients who did not develop de novo DSA (3.7 vs. 5.3 log_10_ cp/mL, respectively, *p* = 0.0023). No differences were found in eGFR at the end of follow-up between patients with and without de novo DSAs (64.00 vs. 58.00 mL/min/1.73 m^2^, respectively, *p* = 0.1716). However, the albumin/creatinine ratio was higher for patients who developed de novo DSAs after KT than for those who did not (63.80 vs. 16.40 mg/g, respectively, *p* = 0.0085).

## 4. Discussion

In the present study, 81 KT patients were prospectively followed in the first year after KT. A robust association was observed between early TTV DNA kinetics and post-transplant infection. We found that the detection of an infectious event in the first year after KT was more probable among patients with higher increases in TTV viral load between the 1st week and the 1st month after KT. The optimal cut-off value of TTV viral load variation between these time points which best discriminates patients with and without infection was 2.65 log_10_ cp/mL. Furthermore, a TTV viral load variation higher than 2.65 log_10_ cp/mL has a four-fold increase in the odds of development of infection after KT. As a marker, this has a high sensitivity of 99.73%; thus, we suggest that this cut-off is optimal to discriminate patients who will develop an infection. Specificity was also high (83.67%); as such, it is expected that this cut-off value is also reliable to identify patients who will be free of infection in the first year after KT. Fernández-Ruiz et al. [20] prospectively analyzed a cohort of 221 KT patients and also found that TTV DNA loads at month 1 were higher among patients who subsequently developed post-transplant infection. Authors stated that TTV DNA load above 3.14 and 4.56 log_10_ cp/mL at month 1 predicted the occurrence of post-transplant infection with an adjusted hazard ratio of 2.88. Both studies validate early TTV kinetics as a useful tool for the prediction of infection in the first year after KT. Furthermore, in a previous study, we demonstrated that TTV viral load predicts the response to SARS-CoV-2 vaccination in KT patients. These studies support the hypothesis that TTV load is an effective surrogate marker of immune competency [21].

Our work also presents a novel revelation of an association between TTV DNA viral load and the presence of DSAs after KT. We found that patients with preformed DSAs had higher TTV DNA viral loads at the 6th and 9th months after KT, compared to patients without preformed DSAs. We postulate that these results reflect a state of higher immunosuppression in patients with preformed DSAs, since all of them received thymoglobulin as part of induction therapy and maintained higher tacrolimus levels throughout the first year after KT, compared to patients who did not have DSAs.

Regarding the formation of de novo DSAs during the first year after KT, we found that patients who developed this type of antibody had lower TTV viral loads in the 1st year after KT. Schiemann et al. [22] assessed the association between TTV load in the peripheral blood and antibody-mediated rejection in a cross-sectional study of 86 recipients with DSAs. They found that the TTV load in patients with DSAs and antibody-mediated rejection (ABMR) was by far lower than in recipients with DSAs but without ABMR. Additionally, authors found that higher TTV levels were associated with a decreased risk for ABMR. Some underestimation of ABMR prevalence may be present in our cohort, as we did not perform protocol biopsies. However, assuming that the formation of de novo DSAs preclude the development of ABMR, our results are in line with the authors.

In summary, lower TTV viral loads reflect low levels of immunosuppression, leading to formation of new DSAs. This finding reinforces an eventual role of TTV viral load as a potential surrogate marker for humoral immunity, and a role of TTV kinetics and TTV viral load in the prediction of post-transplant functional immunity. We had only four biopsy-proven rejection episodes, which was insufficient to evaluate the potential role of TTV viral load as a predictor of rejection episodes.

Our work can also provide additional insight into the prevalence of TTV among end-stage renal disease patients. In our study, 72.5% of patients had detectable TTV DNA at the time of KT. Prevalence of detectable TTV viremia among patients on hemodialysis has been estimated between 41.7% and 80.0% [6], higher than in healthy individuals, supporting the hypothesis that TTV replication is enhanced by end-stage renal disease-associated impaired immune competency and by persistent low-grade inflammatory status [23,24]. Additionally, we report a median pre-transplant TTV DNA level of 3.10 log_10_ cp/mL, consistent with previously published data, where TTV DNA loads have been consistently reported in the range of 2.9–4.4 log_10_ cp/mL [6].

Abbas et al. [25] demonstrated the development of a bidirectional transfer of anelloviruses between the graft and the host. As suggested by Forqué et al. [26], it is plausible that the observed increase in TTV viral load between the pre- and post-transplantation periods may be partially explained by the transfer of donor’s TTV within the graft. Thus, as TTV viral load was not evaluated in donors and genotyping of TTV was not performed, we cannot differentiate between reactivation and graft-mediated infection. Evidence that TTV viral load increase is solely a reactivation of a quiescent endogenous virus can only be obtained among individuals who received immunosuppression without an associated allograft.

Our study validates previously published data on post-transplant TTV kinetics. We noted a progressive increase in TTV viral load from baseline to a peak at month 3 and a slight decrease after month 6, when immunosuppression reaches a lower plateau dose. It is already established that, once immunosuppression is initiated, TTV DNA viral loads rise during the first weeks to months and a peak is observed in most cases by month 3 [18,19], thereafter reaching a steady-state phase with stabilization at levels higher than baseline.

Considering the presumed role of TTV as a surrogate marker of immune status, viral replication kinetics during the post-transplant period is mainly dictated by the amount and type of immunosuppression. Induction therapy with lymphocyte-depleting agents has been associated with higher TTV viral load [27,28]. We were unable to observe such an association in our cohort. One possible explanation relies on the fact that most of our cohort’s patients received only 3 to 5 days of thymoglobulin with low cumulative induction doses. Furthermore, the small number of patients included in the analysis precluded more solid conclusions with regard to the role of immunosuppression in TTV replication.

Measurement of serum immunoglobulin is a widely available surrogate for the functional status of humoral immunity. The presence of post-transplant hypogammaglobulinemia (HGG) has been shown to be associated with an increased risk of infection after KT [10]. The rates of IgG after KT ranges from 45% [29] to 56% [30], depending on the serum level threshold applied and the timing of monitoring. A meta-analysis that included more than 600 KT recipients reported that the risk of developing overall infection increased by 2.46 times in those with severe IgG HGG [31]. We found that IgG, IgA and IgM abruptly decreased in the first week after KT, recovered somewhat after the first month, but there was no recovery to baseline values before KT. However, we were not able to find an association between HGG and infection at all assessment time points.

Surprisingly, patients with admissions due to infection were more likely to have received IVIg. It is difficult to dissociate the concomitant use of thymoglobulin as numbers are small, but IVIg could also blunt the adaptative humoral response to infection. Furthermore, it was used in patients with higher immunological risk. Unfortunately, we cannot explain this finding and, due to the nature of our data and analysis, causality cannot be inferred.

Post-transplant kinetics of PBLSs may be used as a tool to investigate the functionality of adaptive T-cell responses and the subsequent risk of infection. However, few studies have examined the relationships between PBLSs and infection. Moreover, most studies encompass heterogeneous solid organ transplant populations, and studies that evaluate the role of the depletion of T and, mainly, B cells are scarce. Fernández-Ruiz et al. [32] prospectively measured baseline and post-transplant (months 1 and 6) PBLSs in 304 KT recipients. Authors stated that total lymphocyte, CD3+, CD4+ and CD8+ T cells and NK-cell counts at month 1 were significantly decreased in patients who subsequently developed an opportunistic infection between the 1st and 6th month after KT. The same results were found at month 6, but only for CD8+ T-cells. In our work, at the month 1 assessment, patients with infectious events had lower CD3+ T cells, CD8+ T cells and CD19+ B cells, than did patients who remained free from infection in the first year after KT. Later, between months 3 and 6 after KT, patients who subsequently presented with infection had lower CD3+ T cells and lower CD4+ T cells. According to Dendle et al. [9], there is moderate quality evidence to support monitoring lymphocyte subsets to predict infection, making PBLSs a feasible method of identifying patients at high risk of infection.

The present study has limitations, including its single-center design with the consequent impossibility of external generalizability. The number of recruited patients and infection episodes were low and the follow-up time was relatively short. Additionally, infection episodes only included those where an admission was required. Also, caution is necessary when TTV DNA viremia is linked to infectious complications after KT. Although biologically plausible, data should be analyzed as merely hypothesis-generating. Despite these limitations, our study had several strengths: it was exploratory, prospective and longitudinal in nature and primarily aimed at investigating TTV DNA viral load as a surrogate marker of immunosuppressive status. We were able to find a cut-off value for the variation of TTV DNA viral load with high specificity and sensitivity, suggesting a predictive ability of this parameter to discriminate patients that will suffer infectious complications after KT. It is likely that the availability of larger multicenter studies will refine the predictive value of routine measurements of TTV DNA after KT. Our study generated a novel finding that there is a possible association between lower IgM levels and the presence of DSAs antibodies.

In conclusion, KT recipients are at a high risk of infection. While KT recipients currently present fewer early graft losses and enjoy longer survival, changes in infectious disease risk and immunosuppression remain concerning; thus, increased research is required to develop reliable immune and/or biological markers able to reflect an individual’s level of immunosuppression or immune function. TTV DNA viremia is emerging as a reliable biomarker for the overall state of immunosuppression after SOT. The future availability of international standardization of commercial real-time PCR assays will be crucial to further validate TTV DNA viremia as an effective tool to guide immunosuppression prescription.

## Figures and Tables

**Figure 1 viruses-15-01464-f001:**
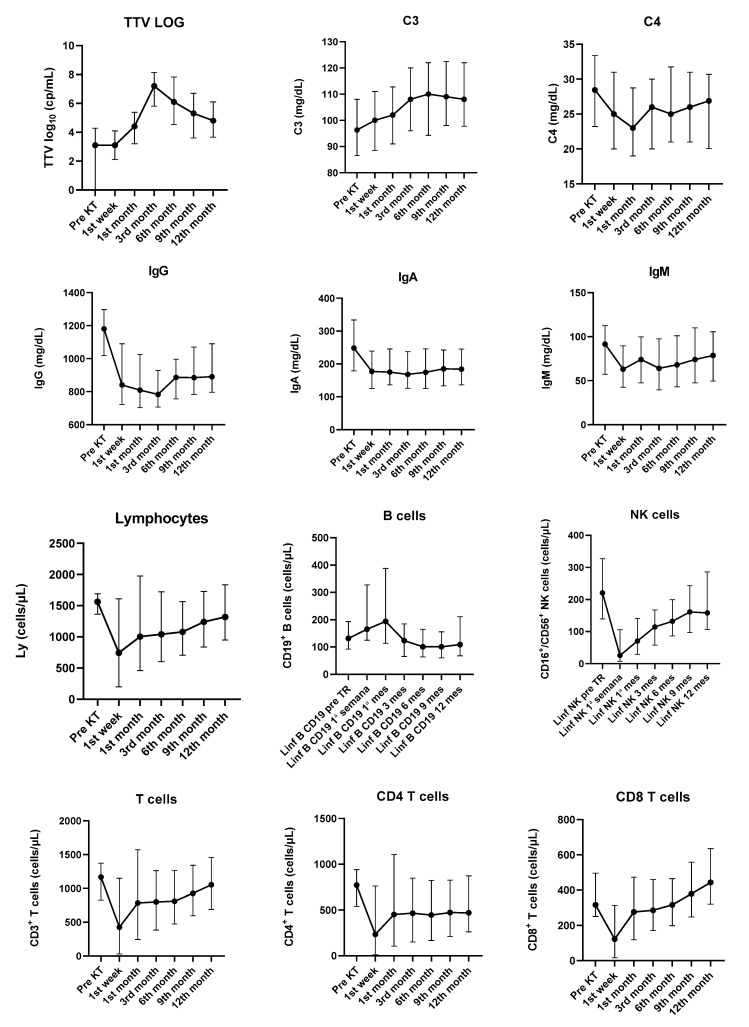
Evolution of TTV viral load (log_10_) and immunological parameters from baseline assessment to the 12th month after transplant. Results are presented as median and interquartile range. KT—kidney transplant.

**Table 1 viruses-15-01464-t001:** Clinical and demographical data of the study population.

	Study Population
Age at transplant, years	
median [IQR]	52.00 [42.50; 61.50]
Gender, male, *n* (%)	
	53 (65.43%)
Dialysis vintage, months	
median [IQR]	63.00 [34.50; 95.00]
Hepatitis C, *n* (%)	5 (6.17%)
Hepatitis B, *n* (%)	1 (1.23%)
HIV, *n* (%)	3 (3.70%)
Type of donor, *n* (%)	
Deceased	70 (86.42%)
Living	11 (13.58%)
Non-heart-beating donor, *n* (%)	77 (95.06%)
Ureteral stent, *n* (%)	64 (79.01%)
Donor age, years	
median [IQR]	56.00 [42.25; 65.00]
Donor gender, male, *n* (%)	39 (48.15%)
Cold ischemia time, hours	
median [IQR]	13.50 [8.25; 17.75]
IgG CMV-positive recipient, *n* (%)	65 (80.25%)
IgG CMV-positive donor, *n* (%)	74 (91.36%)
Delayed graft function, *n* (%)	19 (23.43%)
Diabetes, *n* (%)	
Before KT	10 (12.35%)
NODAT	15 (18.52%)
IMS induction Thymoglobulin, *n* (%)	43 (53.09%)
IMS induction Basiliximab, *n* (%)	38 (46.91%)
IMS induction Rituximab, *n* (%)	3 (3.70%)
IMS induction IVIg, *n* (%)	8 (9.88%)
Maintenance immunosuppression, *n* (%)	
Tacrolimus + mycophenolate mofetil + prednisolone	54 (66.67%)
Tacrolimus + everolimus + prednisolone	25 (30.86%)
Cyclosporine + everolimus + prednisolone	1 (1.23%)
Tacrolimus + prednisolone	1 (1.23%)
Kidney biopsy, *n* (%)	8 (9.88%)
Acute rejection in the 1st year, *n* (%)	4 (4.94%)
PVAN, *n* (%)	
Presumptive	14 (17.28%)
Confirmed	1 (1.23%)
Admissions after KT, *n* (%)	32 (39.51%)
1st admission, Infection, *n* (%)	26/32 (81.25%)
Time from KT to 1st admission, months,	
median [IQR]	2.6 [1.1; 5.2]
2nd admission, Infection, *n* (%)	
	13/32 (40.63%) *
Time from KT to 2st admission, months,	
median [IQR]	4.8 [2.5; 6.7]
3rd admission, Infection, *n* (%)	3/13 (23.08%) **
Time from KT to 3rd admission, months,	
median [IQR]	8.0 [2.5; 8.8]

IQR: interquartile range, KT: kidney transplant; NODAT: new onset diabetes after transplantation; PVAN: polyomavirus nephropathy; IVIG: Intravenous immune globulin; CMV: cytomegalovirus; HIV: human immunodeficiency virus; IMS: immunosuppression. * 2/13 patients had non-infectious cause in 1st admission. ** 2 patients had infectious cause in 1st and 2nd admission; 1 had infectious cause in 1st admission only.

**Table 2 viruses-15-01464-t002:** Admissions due to bacterial and viral post-transplant infections.

	*n*	%
Acute pyelonephritis with or without bacteriemia		
*Klebsiella pneumoniae*	9	23.69
*Escherichia coli*	8	21.05
*Enterococcus faecalis*	1	2.63
Without microbial identification	1	2.63
Pneumonia	3	7.90
CMV infection/disease	2	5.26
COVID-19	7	18.42
Herpes zoster	2	5.26
Acute gastroenteritis	1	2.63
Acute cholecystitis	1	2.63
Febrile neutropenia without microbial identification	3	7.90

CMV: cytomegalovirus; COVID-19: coronavirus disease 2019.

**Table 3 viruses-15-01464-t003:** Clinical and laboratory parameters in patients admitted with infections and patients without admissions (*n* = 75).

	Patients with Admissions *n* = 26	Patients without Admissions *n* = 49	*p*-Value
Age at transplant, years,	54 [45; 63.50]	49 [43; 62]	0.4234
median [IQR]			(MW)
Gender, male, *n* (%)	16 (61.54)	31 (63.27)	>0.9999
			(Fisher)
Previous KT (*n*/%)	4 (15.38)	6 (12.24)	0.7306
			(Fisher)
Dialysis vintage, months,	82 [46;109]	63 [37; 95]	0.3419
median, IQR			(MW)
Type of TSFR			
Hemodialysis	24 (92.31)	39 (79.59)	0.3525
Peritoneal dialysis	1 (3.85)	6 (12.24)	(Chi-Sq)
Pre emptive	1 (3.85)	4 (81.63)	
Hepatitis C (*n*/%)	2 (7.69)	2 (4.08)	0.6059
			(Fisher)
Hepatitis B (*n*/%)	0 (0)	1 (2.04)	>0.9999
			(Fisher)
HIV (*n*/%)	2 (7.69)	0 (0)	0.1171
			(Fisher)
Type of donor (*n*/%)			
Deceased	24 (92.31)	40 (81.63)	0.3106
Living	2 (7.69)	9 (18.37)	(Fisher)
Non heart beating donor (*n*/%)	1 (3.85)	3 (6.12)	>0.9999
			(Fisher)
Ureteral stent (*n*/%)	6 (23.08)	9 (18.37)	0.7629
			(Fisher)
Donor age, years (median, IQR)	*n* = 25		0.1631
	58 [46; 67]	52 [38; 63]	(MW)
Donor gender, male (*n*/%)	14 (53.85)	25 (51.02)	>0.9999
			(Fisher)
Cold ischemia time, hours (median, IQR)	13 [10; 18]	13 [7; 17]	0.5628
			(MW)
IgG CMV-positive recipient, (*n*/%)	22 (84.62)	37 (75.51)	0.5546
			(Fisher)
IgG CMV-positive donor, (*n*/%)	25 (96.15)	44 (89.80)	0.6580
			(Fisher)
Delayed graft function (*n*/%)	5 (19.23)	11 (22.45)	>0.9999
			(Fisher)
Diabetes (*n*/%)			
Before KT	4 (15.38)	6 (12.24)	0.3328
NODAT	7 (26.92)	7 (14.29)	(Chi-Sq)
IMS induction Thymoglobulin (*n*/%)	14 (53.85)	24 (48.98)	0.8093
			(Fisher)
IMS induction Basiliximab (*n*/%)	12 (46.15)	25 (51.02)	0.8093
			(Fisher)
IMS induction Rituximab (*n*/%)	1 (3.85)	2 (4.08)	>0.9999
			(Fisher)
IMS induction IVIg (*n*/%)	3 (11.54)	5 (10.20)	>0.9999
			(Fisher)
Maintenance immunosuppression (*n*/%)			
Tacrolimus + MMF + prednisolone	15 (57.69)	37 (75.51)	0.1598
Tacrolimus + everolimus + prednisolone	9 (34.62)	12 (24.49)	(Chi-Sq)
Cyclosporine + everolimus + prednisolone	1 (3.85)	0 (0)	
Tacrolimus + prednisolone	1 (3.85)	0 (0)	
Kidney biopsy (*n*/%)	2 (7.69)	1 (2.04)	0.2743
			(Fisher)
Acute rejection in the 1st year (*n*/%)	2 (7.69)	0 (0)	0.1171
			(Fisher)
PVAN (*n*/%)			
Presumptive	2 (7.69)	9 (18.37)	0.3367
Confirmed	0 (0)	1 (2.04)	(Chi-Sq)

IQR: interquartile range, KT: kidney transplant; NODAT: new onset diabetes after transplantation; PVAN: polyomavirus nephropathy; IVIG: Intravenous immune globulin; CMV: cytomegalovirus; HIV: human immunodeficiency virus; IMS: immunosuppression.

**Table 4 viruses-15-01464-t004:** Dynamics of immune and microbiologic parameters along the study period for patients admitted due to infectious complications compared to patients without.

With admission (*n* = 26) vs. Without Admission (*n* = 49)	PRE-TR	1st WEEK	1st MONTH	3rd MONTH	6th MONTH	9th MONTH	12th MONTH	*p*-Value
TTV, cp/mL	5832	2852	78,004	11,099,956	1,700,097	381,132	191,166	0.2238 ***
median [IQR]	[380; 24,608]	[111; 12,320]	[5661; 403,278]	[52,600; 91,372,336]	[46,826; 16,316,084]	[3688; 11,321,619]	[8416; 992,170]	
With	840	921	24,130	9,510,760	523,694	161,321	37,917	
Without	[0; 12,949]	[127; 13,241]	[868; 171,058]	[594,792; 120,119,216]	[20,891; 85,892,567]	[406; 3,873,827]	[1992; 2,833,886]	0.3925 #
*p*-value *	0.2095	0.4717	0.1213	0.5973	0.8448	0.5300	0.4734	
Log_10_ TTV, cp/mL	3.80	3.45	4.90	7.00	6.20	5.60	5.30	<0.0001 ***
median, IQR	[2.50; 4.35]	[1.95; 4.10]	[3.70; 5.60]	[4.70; 7.95]	[4.65; 7.20]	[3.55; 7.10]	[3.90; 5.93]	
With	2.90	3.00	4.40	7.00	5.70	5.20	4.60	<0.0001 #
Without	[0.00; 4.10]	[2.10; 4.15]	[2.95; 5.25]	[5.75; 8.05]	[4.30; 7.95]	[3.60; 6.60]	[3.30; 6.40]	
*p*-value *	0.2029	0.5100	0.1224	0.5971	0.8804	0.5117	0.4463	
Complement C3, mg/dL	93.3	96.0	102.0	111.0	104.0	109.0	110.5	<0.0001 ***
median, IQR	[81.9; 108.5]	[85.5; 109.3]	[91.0; 115]	[99.5; 124.8]	[90.0; 137.0]	[97.3; 123.8]	[97.5; 128.3]	
With	96.3	100.0	101	104.0	106.0	107.0	106.0	<0.0001 #
Without	[85.9; 108.0]	[89.0; 112.0]	[88; 111]	[95.0; 118.0]	[95.5; 117.5]	[97.0; 120.0]	[97.1; 119.5]	
*p*-value *	0.5567	0.4333	0.5117	0.3362	0.7313	0.5933	0.3419	
Complement C4, mg/dL	27.7	24.5	24.0	26.5	24.0	27.0	27.3	0.3951 ***
median, IQR	[22.1; 35.7]	[19.0; 31.3]	[18.0; 32.5]	[20.0; 29.3]	[20.5; 33.0]	[21.8; 30.3]	[19.7; 32.3]	
With	28.4	25.0	22.0	25.0	24.0	25.0	25.4	<0.0001 #
Without	[24.0; 33.0]	[20.0; 31.5]	[19.0; 27.0]	[19.5; 30.0]	[20.5; 29.0]	[19.5; 30.5]	[18.1; 29.7]	
*p*-value *	0.7788	0.8572	0.1384	0.7251	0.6108	0.4974	0.4633	
IgG, mg/dL	1260	918	863	824	916	972	946	0.0001 ***
median, IQR	[1095; 1510]	[690; 1123]	[709; 1055]	[685; 1023]	[741; 1110]	[807; 1133]	[801; 1160]	
With	1140	822	780	771	877	862	884	0.0001 #
Without	[982; 1245]	[726; 1075]	[700; 999]	[714; 875]	[767; 945]	[772; 982]	[776; 1065]	
*p*-value *	0.0410	0.7131	0.3238	0.2885	0.5081	0.1230	0.1950	
IgA, mg/dL	245	176	177	162	164	185	183	<0.0001 ***
median, IQR	[178; 329]	[111; 215]	[134; 218]	[124; 200]	[126; 209]	[137; 2079]	[148; 224]	
With	259	178	176	173	188	188	196	<0.0001 #
Without	[194; 336]	[133; 244]	[139; 256]	[130; 257]	[130: 258]	[130; 248]	[136; 258]	
*p*-value *	0.5798	0.3119	0.3351	0.1747	0.1373	0.4078	0.4269	
IgM, mg/dL	97.00	65.00	65.00	85.00	85.00	86.50	90.20	0.0459 ***
median, IQR	[60.50; 120.50]	[44.00; 91.50]	[44.00; 125.00]	[34.50; 118.00]	[40.50; 109.50]	[50.50; 133.50]	[55.38; 111.80]	
With	87.00	55.00	75.00	61.00	67.00	66.00	71.70	<0.0001 #
Without	[83.50; 110.50]	[42.50; 82.50]	[47.00; 95.00]	[41.50; 89.50]	[43.00; 95.00]	[44.00; 96.00]	[49.15; 104.50]	
*p*-value *	0.4279	0.3064	0.6599	0.2233	0.3986	0.0990	0.2068	
BKPyV viremia								
With	-							
Pos, *n* (%)		0	1	3	1	2	2	0.5818
Neg, *n* (%)	-	26	24	23	24	24	24	CS
Without								
Pos, *n* (%)		0	0	7	5	5	7	0.0147
Neg, *n* (%)		49	49	42	44	44	42	CS
*p*-value **		1.000	0.3378	1.000	0.6569	1.000	0.4835	
JCPyV viremia	-							
With								
Pos, *n* (%)	-	0	0	1	1	1	1	0.8443
Neg, *n* (%)		26	25	25	24	25	25	CS
Without								
Pos, *n* (%)		0	0	0	1	2	2	0.3160
Neg, *n* (%)		49	49	49	48	47	47	CS
*p*-value **		1.000	1.000	0.3467	1.000	1.000	1.000	
BKPyV viruria	-							
With								
Pos, *n* (%)	-	3	5	8	7	8	6	0.5921
Neg, *n* (%)		22	20	18	18	18	20	CS
Without								
Pos, *n* (%)		3	5	11	15	13	13	0.0088
Neg, *n* (%)		48	44	37	34	36	36	CS
*p*-value **		0.3884	0.2904	0.5786	1.000	0.7888	0.7884	
JCPyV viruria	-							
With								
Pos, *n* (%)	-	4	5	5	6	5	10	0.4438
Neg, *n* (%)		21	20	21	19	21	16	CS
Without								
Pos, *n* (%)		10	8	11	13	13	14	0.7324
Neg, *n* (%)		38	41	37	36	36	35	CS
*p*-value **		0.7590	0.7517	0.7761	1.000	0.5774	0.4403	
Creatinine, mg/dL	-	1.70	1.49	1.44	1.37	1.48	1.34	0.0079 ***
median, IQR		[1.33; 3.53]	[1.05; 1.95]	[0.94; 1.78]	[1.16; 1.81]	[1.04; 1.86]	[1.10; 1.91]	
With	-	1.70	1.33	1.24	1.29	1.35	1.23	0.0002 #
Without		[1.20; 2.45]	[1.08; 1.79]	[1.08; 1.54]	[1.09; 1.48]	[1.06; 1.59]	[1.07; 1.63]	
*p*-value *		0.6840	0.4050	0.5483	0.2307	0.2233	0.4735	
eGFR, mL/min/1.73m^2^	-	37.50	50.00	55.00	54.00	52.00	57.50	<0.0001 ***
median, IQR		[18.50; 59.75]	[36.00; 70.00]	[43.00; 75.00]	[38.50; 61.00]	[40.25; 65.75]	[37.50; 67.75]	
With	-	42.00	56.00	58.00	60.00	61.00	61.00	<0.0001 #
Without		[26.00; 61.00]	[44.50; 69.00]	[48.00; 72.50]	[48.50; 75.009]	[45.00; 76.00]	[45.00; 70.50]	
*p*-value *		0.5557	0.3954	0.4838	0.1531	0.1401	0.3198	
Albumin creatinine ratio, mg/g	-	-	49.40	37.00	42.10	33.65	21.70	0.2834 ***
median, IQR			[20.60; 108.50]	[10.60; 71.45]	[14.20; 86.10]	[9.38; 87.15]	[13.45; 70.50]	
With	-	-	26.20	18.40	18.70	16.90	17.10	0.4062 #
Without			[12.43; 72.75]	[9.00; 37.75]	[8.10; 88.75]	[7.95; 63.40]	[7.89; 45.30]	
*p*-value *			0.1710	0.0780	0.2992	0.2453	0.1970	
Tacrolimus, µg/mL	-	7.15	9.00	8.65	6.55	7.20	5.85	0.0076 ***
median, IQR		[5.75; 11.78]	[7.70; 10.70]	[5.55; 9.25]	[5.05; 8.68]	[5.10; 8.50]	[4.90; 6.83]	
With	-	7.50	10.40	8.70	7.50	7.30	6.50	<0.0001 ***
Without		[6.00; 9.00]	[8.70; 12.15]	[7.25; 9.90]	[6.30; 8.70]	[6.20; 8.60]	[5.55; 7.95]	
*p*-value *		0.6196	0.0970	0.5298	0.1012	0.6847	0.0576	
CRP, mg/dL	-	0.76	0.14	0.30	0.15	0.22	0.16	0.5807 ***
median, IQR		[0.51; 1.18]	[0.10; 1.09]	[0.10; 1.83]	[0.10; 0.45]	[0.10; 0.46]	[0.10; 0.32]	
With	-	0.75	0.10	0.10	0.14	0.15	0.12	<0.0001 ***
Without		[0.36; 1.54]	[0.10; 0.13]	[0.10; 0.30]	[0.10; 0.39]	[0.10; 0.46]	[0.10; 0.39]	
*p*-value *		0.9010	0.0078	0.0114	0.7334	0.4007	0.8605	
WBC, Cells/uL	5900	6950	6100	4350	4600	5150	5500	<0.0001 ***
median, IQR	[4750; 7100]	[4950; 9325]	[3850; 8150]	[2975; 5725]	[3600; 6500]	[3650; 7350]	[4250; 7950]	
With	6600	7200	6800	5000	5400	5600	5700	<0.0001 #
Without	[5300; 8100]	[5500; 9400]	[5000; 8400]	[4050; 6600]	[4300; 6600]	[4700; 6950]	[4800; 6700]	
*p*-value *	0.0625	0.6600	0.1434	0.0663	0.2261	0.3591	0.7715	
Total lymph, Cells/uL	1698	427	750	833	936	954	1292	0.0014 ***
median, IQR	[1363; 2810]	[208; 1217]	[422; 1195]	[464; 1282]	[742; 1353]	[794; 1751]	[978; 2115]	
With	1550	1007	1529	1226	1350	1287	1358	0.0176 ***
Without	[1315; 1654]	[210; 1658]	[517; 2593]	[685; 1890]	[747; 1776]	[925; 1797]	[926; 1833]	
*p*-value *	0.1977	0.3324	0.0327	0.0463	0.2296	0.2809	0.5013	
CD3+ T cells, Cells/uL	1214	230	570	629	690	757	979	0.0014 ***
median, IQR	[814; 2290]	[44; 799]	[209; 987]	[394; 969]	[477; 1011]	[508; 1339]	[735; 1603]	
With	1166	588	1093	981	994	952	1064	0.0227 ***
Without	[827; 1319]	[43; 1229]	[417; 1947]	[482; 1464]	[527; 1395]	[672; 1374]	[675; 1439]	
*p*-value *	0.4604	0.3555	0.0359	0.0595	0.2184	0.3740	0.5641	
CD4+ T cells, Cells/uL	808	99	247	334	359	423	464	0.0103 ***
median, IQR	[539; 1357]	[17; 597]	[79; 660]	[113; 640]	[161; 640]	[157; 780]	[222; 908]	
With	770	405	686	619	605	563	571	0.0193 ***
Without	[581; 930]	[17; 892]	[210; 1424]	[198; 1046]	[248; 957]	[281; 899]	[276; 884]	
*p*-value *	0.6714	0.3408	0.0246	0.0280	0.1002	0.1286	0.5833	
CD8+ T cells Cells/uL	333	120	175	210	290	358	462	<0.0001 ***
median, IQR	[272; 885]	[21; 203]	[101; 382]	[137; 438]	[216; 453]	[224; 658]	[328; 684]	
With	313	217	363	309	361	384	437	<0.0001 ***
Without	[216; 407]	[25; 352]	[150; 564]	[201; 486]	[196; 477]	[274; 552]	[296; 574]	
*p*-value *	0.2776	0.1669	0.0526	0.1390	0.6989	0.8879	0.1932	
CD19+ B cells Cells/uL	149	155	153	87	93	91	104	0.0064 ***
median, IQR	[117; 239]	[127; 236]	[79; 230]	[52; 137]	[61; 158]	[50; 128]	[61; 237]	
With	132	174	234	136	103	106	123	<0.0001 ***
Without	[93; 187]	[120; 346]	[144; 435]	[91; 209]	[60; 174]	[66; 168]	[78; 207]	
*p*-value *	0.4815	0.5644	0.0290	0.0154	0.5655	0.2213	0.4223	
NK cells Cells/uL	279	19	66	98	126	166	187	<0.0001 ***
median, IQR	[159; 367]	[10; 82]	[28; 132]	[48; 161]	[88; 195]	[93; 202]	[115; 344]	
With	185	43.00	72	124	132	159	143	<0.0001 ***
Without	[120; 315]	[6; 115]	[34; 151]	[58; 173]	[79; 198]	[98; 250]	[102; 214]	
*p*-value *	0.4043	0.6312	0.5377	0.3336	0.9241	0.7131	0.2703	

* Mann–Whitney U Test; ** Fisher Exact Test; CS Chi-Square test; *** Mixed Effects Analysis with the Geisser–Grenhouse correction; # RM one-way ANOVA, with the Geisser–-Grenhouse correction.

**Table 5 viruses-15-01464-t005:** Clinical and laboratory parameters in patients with and without de novo donor-specific antibodies.

Patients without Pre-Formed DSAs (*n* = 61)	Patients with De Novo DSA *n* = 11	Patients without De Novo DSA *n* = 50	*p*-Value
Age at transplant, years,			0.8568
median [IQR]	47 [34; 66]	50 [42; 61]	(MW)
Gender, male, *n* (%)			0.3017
	9 (81.82%)	31 (62.00%)	(Fisher)
Dialysis vintage, months,			0.6334
median [IQR]	56 [37; 92]	63 [32; 94]	(MW)
Hepatitis C, *n* (%)			>0.9999
	0 (0%)	4 (8.00%)	(Fisher)
Hepatitis B, *n* (%)			>0.9999
	0 (0%)	0 (0%)	(Fisher)
HIV, *n* (%)			0.3306
	1 (9.09%)	1 (2.00%)	(Fisher)
Type of donor, *n* (%)			
Deceased	10 (90.91%)	42 (84.00%)	>0.9999
Living	1 (9.09%)	8 (16.00%)	(Fisher)
Non-heart-beating donor, *n* (%)			>0.9999
	0 (0%)	3 (6.00%)	(Fisher)
Ureteral stent, *n* (%)			0.4569
	4 (36.36%)	12 (24.00%)	(Fisher)
Donor age, years			0.0485
median [IQR]	45 [36; 59]	61 [47; 67]	(MW)
Donor gender, male, *n* (%)			0.7396
	6 (54.55%)	22 (44.00%)	(Fisher)
Cold ischemia time, hours,			0.6273
median [IQR]	12 [8; 15]	14 [8; 18]	(MW)
IgG CMV-positive recipient, *n* (%)			0.4290
	10 (90.91%)	38 (76.00%)	(Fisher)
IgG CMV-positive donor, *n* (%)			0.2941
	9 (81.82%)	46 (92.00%)	(Fisher)
Delayed graft function, *n* (%)			>0.9999
	2 (18.18%)	11 (22.00%)	(Fisher)
Diabetes, *n* (%)			
Before KT	3 (27.27%)	7 (14.00%)	0.5574
NODAT	2 (18.18%)	10 (20.00%)	(Chi-Sq)
IMS induction Thymoglobulin (*n*/%)			0.5258
	4 (36.36%)	24 (48.00%)	(Fisher)
IMS induction Basiliximab (*n*/%)			0.5258
	7 (63.64%)	26 (52.00%)	(Fisher)
IMS induction Rituximab (*n*/%)			0.3306
	1 (9.09%)	1 (2.00%)	(Fisher)
IMS induction IVIg, *n* (%)			>0.9999
	0 (0%)	1 (2.00%)	(Fisher)
Maintenance immunosuppression, *n* (%)			
Tacrolimus + MMF + prednisolone	7 (63.64%)	33 (66.00%)	0.1841
Tacrolimus + everolimus + prednisolone	3 (27.27%)	16 (32.00%)	(Chi-Sq)
Cyclosporine + everolimus + prednisolone	1 (9.09%)	0 (0%)	
Tacrolimus + prednisolone	0 (0%)	1 (2.00%)	
Kidney biopsy (*n*/%)			0.4554
	1 (9.09%)	2 (4.00%)	(Fisher)
Acute rejection in the 1st year, *n* (%)			0.3306
	1 (9.09%)	1 (2.00%)	(Fisher)
PVAN (*n*/%)			
Presumptive	2 (18.18%)	10 (20.00%)	>0.9999
Confirmed	0	0	(Fisher)
De novo DSAs	TOTAL (Class I/Class II)	-	
1 Month	2 (0/2)		
3 Month	3 (0/3)		
6 Month	5 (2/3)		
9 Month	7 (2/6) *		
12 Month	6 (1/5) *		

IQR: interquartile range, KT: kidney transplant; NODAT: new onset diabetes after transplantation; PVAN: polyomavirus nephropathy; IVIG: Intravenous immune globulin; CMV: cytomegalovirus; HIV: human immunodeficiency virus; IMS: immunosuppression. * 1 patient was positive for both anti-MHC class I and anti-MHC class II DSAs.

## Data Availability

Data is unavailable due to privacy or ethical restrictions.

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
