# Peer review of "Kinetics of Torque Teno Virus Viral Load Is Associated with Infection and De Novo Donor Specific Antibodies in the First Year after Kidney Transplantation: A Prospective Cohort Study"

_viruses, 2023, doi:10.3390/v15071464_

Round 1

Reviewer 1 Report (Previous Reviewer 2)

The author have substantially improved the text. Now, according this reviewer, the manjuscript is available for pubblication.

Reviewer 2 Report (Previous Reviewer 1)

The paper is well written and the modifications provided increase the clarity of the paper

This manuscript is a resubmission of an earlier submission. The following is a list of the peer review reports and author responses from that submission.

Round 1

Reviewer 1 Report

In their paper, Querido and coll, reported the prognostic values of TTV in renal transplant recipient for infections as well as acute rejection. Despite it is a small study, they observed a correlation with these both markers confirming previous results. Interestingly, they describe the initial TTV replication level in patients at the time of transplantation. It would be interesting to precise a possible correlation with previous immunosuppressive treatment (for example for vasculitis) as well any correlation with the age or duration of dialysis. They also found that the variation of the viral load is a more robust marker of the risk of infection.

The work is well done but the link between the pre transplantation period and the initial positivity has to be developed.

Minor point: Table 4: the number of patients in the table is not corresponding to the number of patients included in the study. Have the authors some explanations (75 vs 81)

NA

Reviewer 2 Report

Sara Querido and colleagues reported TTV viral load in 81 kidney transplant (KT) patients at different time along the first year post-transplant to address overall kinetics and the relationship with deleterious events, including episodes of infection and the formation of de novo donor specific antibodies (DSAs). Overall, 26 patients (32.1%) resented a total of 38 infection episodes post-KT. Among patients examined, patients with infectious events had statistically lower T-cells, CD8+ T-cells and B-cells compared to infection-free patients in the first month post-KT. None difference in the immunosuppression therapy was not associated with more 48 infections. Higher increases in TTV viral loads between 1st wk-1st mo post-KT with patients who developed de novo DSAs had lower TTV DNA viral loads in the 1st year after KT comparing with patients who did not develop DSA. Authors claimed that TTV viremia is a promising strategy for defining infectious risk in the 1st year post-KT. The study is well done and reported several relevant data on the argument. However, some points should be implemented to improve the study.

Main points

1- Introduction: a brief description of the different genotype and species of TTV could be introduced to shed light on the variability of the anellovirus.

2- Material and Methods: a description of the primer and probe used in the different PCR used to detect JCPyV, BKpyV and other virus investigated could be added. Moreover, the amount of plasma/serum/urine used to extract DNA should be indicated.

3- Material and Methods: a more description of the assay to detect immunological response should be added.

4- The increase TTV viral load is related to a new infection or to a virus reactivation during the immunosuppression? Can the authors give a potential explanation?

5-The different TTV viral load outcome between the firs 6 months and after should be comment.

6- The presence of different genogroup or species of TTV could be relevant in the different viremia outcome after transplantation. Have the authors these data? If not, the authors should give a potential comment in the discussion about this.

7- The different viral load observed with JCPyV and BKPyV should be briefly comment in the text.

Minor points

1- typos should be corrected

2- BKV and JCV should be changed in BKPyV and JCPyV

Reviewer 3 Report

In this manuscript, Querido and colleagues investigate the kinetics of TTV viral load in kidney transplant recipients, notably to predict infection and de novo donor specific antibodies. This topic is of great interest in transplantation. They present interesting data albeit in a slightly convoluted way with some information/analyses that may be missing or imprecise.

First, the definition of infection as requiring 1) antimicrobial or antiviral therapy AND 2) in an hospitalization setting may pose some problems such as excluding a large amount of infectious events (e.g. CMV reactivation requiring treatment but not necessarily in-hospital) ; moreover, according to this Polyomavirus infections should not have been considered (no antiviral therapy is given but rather immunosuppression tapering)

Sample matrices are not clear : was TTV measured in serum (line 100) or plasma (line 150) ? Was CMV measured in blood (line 129) or plasma (line 176) ?

The « threshold for defining positivity » for viral PCR is an unclear notion. Is this the sensitivity limit of the method for quantification/detection (i.e. no lower amount is ever reached by the technique) or is it an established threshold (and how ?) meaning that positive values with loads lower than this are considered negative ?

What is the meaning of « no changes in the kinetics of JCV, BKV, CMV » ? Does it only mean that the number of patients replicating these viruses are not significant ? And are all replication events taken into consideration or only those requiring antiviral treatment in hospitalization ?

Results sections 3.4, 3.5, 3.6 and 3.7 could be combined and rearranged since many other analyses could be performed. Section 3.7 adresses TTV at specific time points (between week 1 and month 1) and should be included in the paragraphs adressing these timepoints. Furthermore TTV VL and kinetics at each timepoint should be investigated more thoroughly :

- What about TTV VL at D0, week 1 (and the variation between these timepoints) and infections until 1st month (paragraph 3.4) ?

- What about TTV VL at D0, week 1, month 1 (and the variation between D0 and week 1, D0 and month 1) and infections until 3rd month (paragraph 3.5) ?

- What about TTV VL at D0, week 1, month 1, month 3 (and the variation between these timepoints) and infections until 6th month (paragraph 3.6) ?

- What about TTV VL at D0 or at week 1, or month 1 (and D0-W1, D0-M1 variation) and infections events during the 1st year ? Why was only the variation W1-M1 investigated ? It would help to compare the results with the litterature, notably with the thresholds cited in the discussion.

Similar remarks for patients with preformed or de novo DSA : what about baseline TTV VL ? And other timepoints ? And VL variations ? Is the meaning of « lower TTV VL in the first year » that all timepoints were grouped in the analysis ? What are the results by timepoint ?

Line 187, TTV VL stabilizes at higher levels, not lower (according to litterature and to the manuscript graph).

There are many massive tables which make the manuscript a bit difficult to grasp. Some tables could be moved to supplementary materials (e.g. Table 3 since most of these data are shown in Figure 1).

Why were rejection cases not investigated in regard to immunological parameters and TTV VL ?

Minor remarks :

There is a problem with the reference numbers (going from xviii to xxi).

Sentences lines 70-73 and 75-75 convey the same meaning, one should be removed or they should be combined.

There is a problem in Discussion line 176 « Error bookmark not defined ».

Line 45 (and another place in the text) : clinical not « clinically events »

Line 159 : the presence « of » BK polyomavirus

Line 225 : Chi square not Qui square

Minor error and spell checks